# WASH, nutrition and health-seeking behavior during COVID-19 lockdowns: Evidence from rural Bangladesh

Margo van Gurp[1]*, Imam M. Riad[2], Kazal A. Islam[2], Md Shariful Islam[3], Remco M. Geervliet[4], Mirjam I. Bakker[1]

**1** KIT Royal Tropical Institute, Amsterdam, The Netherlands, **2** Max Foundation, Dhaka, Bangladesh, **3** Infectious Diseases Division, icddr,b, Dhaka, Bangladesh, **4** Max Foundation, Amsterdam, The Netherlands

* m.v.gurp@kit.nl

**Data Availability Statement:** All survey data files are available from the OSF repository (https://doi.org/10.17605/OSF.IO/9NY5T).

## Abstract

A general lockdown to minimize to slow transmission of COVID-19 in Bangladesh came into effect on March 26th and lasted until May 30th. The lockdown had far-reaching economic implications for the population, with many facing economic hardship due to loss of income. Despite the attempt of the government to ease economic hardship by means of social safety net packages, people suffered from poor access to health services, and financial and food insecurity. This is likely to have disastrous consequences for the nutritional status of young children. This cross-sectional study measured the impact of the first general lockdown on food consumption of young children, access to water, handwashing and health seeking behavior, and the ability to maintain livelihood among households with children under the age of 5, in rural Bangladesh. The result of the analysis suggest that loss of income was reported by almost all respondents across all socio-economic groups. However, the poorest households were less likely to provide for sufficient food for their families and had to reduce consumption of food. Diet diversity and food intake–particularly animal protein sources—for young children were severely affected. On the other, increased awareness of handwashing and access to soap were also reported. The pandemic is likely to be detrimental to the nutritional status of children in Bangladesh and can exacerbate existing health inequities. Strong social safety net programs are needed to protect vulnerable populations to consequences of restrictive measures, supported in design and implementation by non-governmental organizations.

## Introduction

After its emergence in December 2019, COVID-19 swept across the globe, taking millions of lives. At the time of writing this paper, more than 5 million deaths attributed to COVID-19 were reported worldwide [1]. To reduce the community spread of COVID-19 and prevent morbidity and mortality resulting from the disease, countries have implemented a wide range of measures ranging from requirement to wear protective gear in public to full lockdowns [2].

**Funding:** The study was funded by Stichting Max Foundation (20201023). The funder, other than the named authors, had no role in study design, data collection and analysis, decision to publish, or preparation of the manuscript.

**Competing interests:** The authors have declared that no competing interests exist.

Bangladesh's first confirmed case of COVID-19 was reported on March 8[th] 2020 [3], the first death followed on March 18[th]. A general lockdown came into effect on March 26[th] to slow transmission [4]. The public and private sectors were temporarily halted with the exception of emergency services, health care, pharmacies, and essential shops such as bazaars, grocers and department stores. The lockdown was intended to be in place until April 4[th], but eventually ended on May 30[th] after numerous extensions [5]. The main reason for relaxing the lockdown measures was the negative effect it had on the livelihood of the population and to ease the economic hardship imposed, but educational institutions, public transport and public gatherings remained suspended [5].

The pandemic and its preventive measures have far-reaching economic implications for the population of Bangladesh [6]. Informal workers make up 89% of the labor force in Bangladesh [7], and according to the International Labor Organization, they are among the most vulnerable populations and hardest hit by lockdown measures globally [8]. A study conducted by Rahman et al. estimated that 77% of informal workers in Bangladesh with an income above the poverty line fell below the poverty line due to impact of the response to the first wave of COVID-19 [9]. This concurs with findings from an online survey conducted by the Bangladesh Institute of Development Studies, which reported that over 70% of workers in the informal sector experienced months without any income [10]. In addition, respondents with the lowest income (below 15,000 BDT per month) were more likely to experience a reduction or stoppage of income during the pandemic [10, 11].

The impact of the response to COVID-19 on livelihood and food security of people is likely to have disastrous consequences on the nutritional status of children. According to an analysis in the Lancet, an additional 3.9 million South Asian children could suffer from wasting as a result of COVID-19's socio-economic impact [12]. Risk factors for stunting such as food insecurity, poor quality diets, reduced income and limited financial resources, limited care and restricted health services and interrupted education for children and adults are likely to be exacerbated by COVID-19 [11, 13]. A small study conducted in two Bangladeshi cities reports that both the quantity and quality of food are directly affected by loss of income due to COVID-19 restrictions [14]. Affordability of food and accessibility to food markets were the main drivers of reduced food consumption–especially of animal protein sources. Coping mechanisms included food storage, skipping meals and curtailing consumption [14]. This could mean that steady improvements in malnutrition gained by Bangladesh over the past two decades could come to a halt, or worse, be reversed [15, 16].

In an attempt to mitigate the consequences of the COVID-19 crisis, Bangladesh adopted a range of social safety net programs (SSNPs) [17]. Between March and June of 2020 the Bangladeshi government implemented the Gratuit Relief (GR) plan which provided nearly 75 million individuals with food, baby food and/or cash aid. Additional SSNPs followed, such as rice sales at subsidized prices, cash support for informal workers made jobless and cash assistance for workers in the export industry. However, the coverage and timespan of these programs were insufficient to meet the demand of the population [17].

While pharmaceutical interventions such as vaccines and medications are becoming more widely available it is evident that the pandemic is far from over. Additional preventive measures to curb community spread of COVID-19 such as movement restrictions, school closure and closure of non-essential businesses may still be needed from time to time. These interventions are effective in reducing the spread of the virus in communities by reducing mobility, but not without collateral damage [18]. In order to limit the negative consequences of COVID-19 preventive measures–in particular on vulnerable populations—it is imperative to understand if and how communities are affected by them and what long-term risks they might face. Therefore, the primary objective of this study was to explore and describe how livelihood,

access to water, food consumption of young children and health seeking behavior in communities in rural Bangladesh have been affected by the COVID-19 pandemic and preventive measures in 2020. The secondary objective is to assess whether these impacts are different across socio-economic groups. We hypothesize that measures implemented to curb community spread of COVID-19 could have negatively impacted households' ability to provide for their families, subsequently impacting their children's diet. In addition, essential services for maintenance of water supply may have been disrupted resulting in reduced access to safe water. Furthermore, we expect households from lower socio-economic groups to be affected more than wealthier households. The study was commissioned by Max Foundation, which implements a nutrition and WASH intervention program in South Coastal Bangladesh. Results from this study can be used to inform other NGOs working in Bangladesh and government policies to mitigate the negative impact of COVID-19 in an attempt to avoid further exacerbation of health inequities.

## Methods

### Study area and population

This is a cross-sectional study among 407 households with children between the ages of 12 and 59 months old. Respondents were women of at least 18 years old who had at least one child between the ages of 12 and 59 months at the time of the interview. This study has been carried out within the context of the Max Nutri-WASH program, an intervention program implemented in 62 unions across Barguna, Jessore, Khulna, Patuakhali and Sathkira districts (Fig 1). The intervention unions are divided among 5 partner NGOs (PNGO) who are responsible for implementing the program.

### Sampling frame

A sampling frame of households with children under the age of 5 was constructed using a list of households with contact information in the intervention area compiled in 2017–18, and updated with lists of children in the intervention area who participated in growth monitoring sessions in 2019 and 2020. The list of contact information of households was compiled by Max Foundation using a mobile application with the aim to register all households in the intervention area. During the growth monitoring sessions, children's height and weight were measured and their results were recorded in a mobile application. Every child had a unique identifier and could be linked to aforementioned household list. As such, households with young children were extracted from the household list. A total of 275,705 households were listed in 2017–18 in the intervention unions, of which 39,415 were linked to a child under the age of 5. The latter served as the sampling frame of this study.

### Sample size

A stratified random sample of 1955 households was drawn from a list of almost 40,000 households with children under the age of 5. The sample size was calculated based on the following formula:

$$N = [Z^2 p(1-p)]/(C^2 * R)$$

Were $Z$ is the z-statistic, $p$ is the expected proportion, and $C$ is the required level of confidence and $R$ the response rate. Using a power of 80% and a significance level of 5% (Z = 1.96), and assuming a proportion of 0.5 (50%; to obtain the most conservative sample size) and a response rate of 25%. The sample was stratified by PNGO, proportional to stratum size.

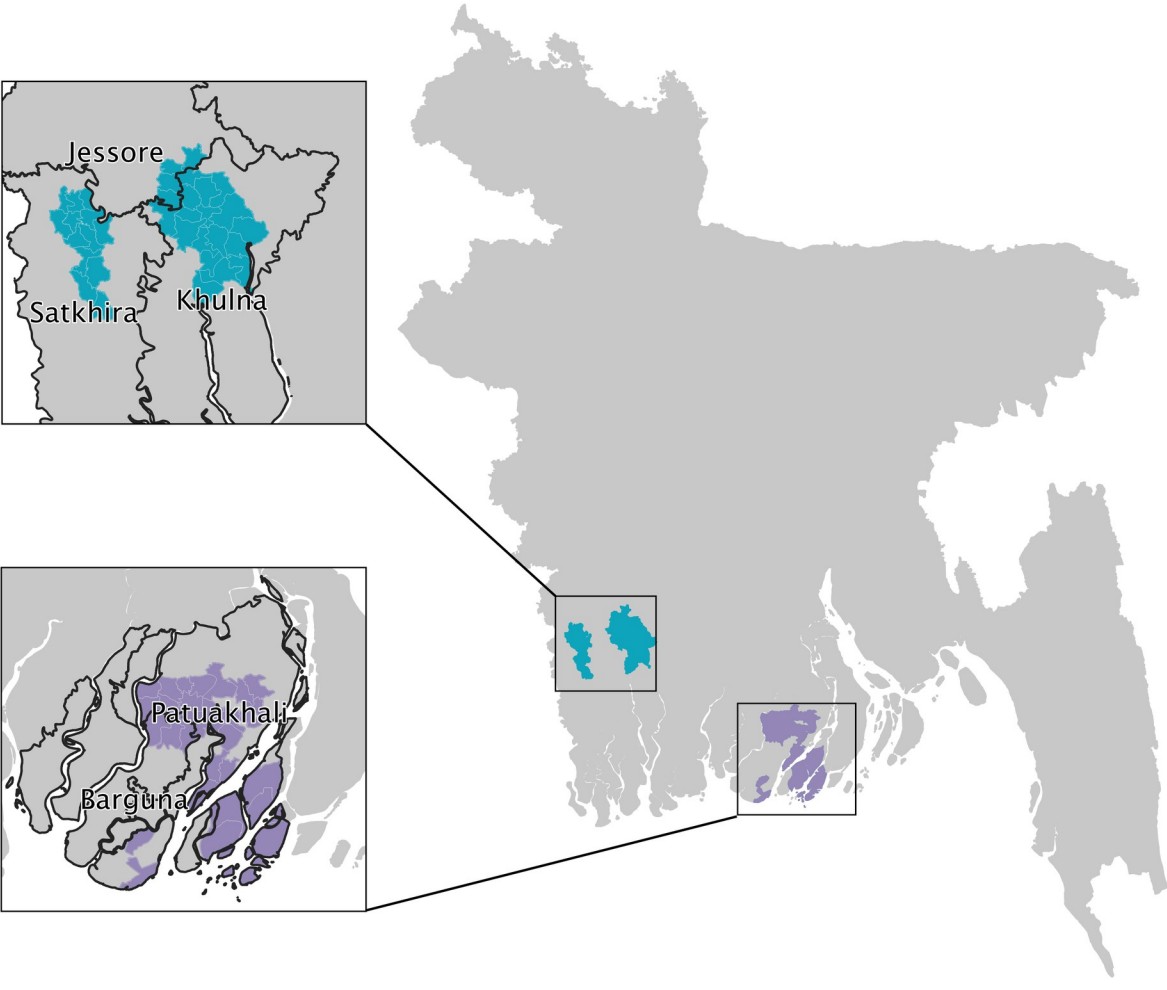

**Fig 1. Study area.**

### Data collection procedures

Phone interviews with beneficiaries were performed in January and February 2021 –approximately 10 months after the start of the first general lockdown–by trained data collectors using an online data entry tool. Data collectors were trained for one day on the questionnaire, online data entry and informed consent procedures, and were experienced in conducting phone interviews. The questionnaire was designed in English and translated into Bangla and piloted prior to data collection. The questionnaire starts with questions pertaining to informed consent and eligibility of the respondent (i.e. age of respondent and children). After consent is obtained and eligibility confirmed, participants were asked questions on how they perceived changes in ability to maintain livelihood, child nutrition, access to growth monitoring services, water and sanitation and health care throughout the pandemic (March 2020 to January 2021). Questions related to child nutrition and growth monitoring were asked in relation to the respondent's youngest child between the ages of 12 and 59 months at the time of the survey. These children were approximately 10 months younger at the start of the general lockdown (I.e between 2 and 49 months). Table 1 provides an overview of the questions described in this study.

Data on the socio-economic status of household was extracted from the household list compiled in 2017–18. Households were classified as poorest, poor, middle-income or rich by

**Table 1. Key outcome variables.**

| Question | Type |
|---|---|
| *Demographic information* | |
| Age of respondent in years | Numerical |
| Number of children under 5 | Numerical |
| Age of youngest child between 12 and 59 months | Numerical |
| Gender of youngest child between 12 and 59 months | Categorical (nominal), single response |
| What is the current main source of household income | Categorical (nominal), multiple response |
| *Livelihood* | |
| Which of the COVID-19 measures has impacted your household the most? | Categorical (nominal), multiple response |
| Has the COVID-19 pandemic impacted your household's ability to maintain livelihoods? | Categorical (ordinal), single response |
| If your household's livelihood has been impacted negatively, how so? | Categorical (nominal), multiple response |
| Which strategies were used by your household to cope with the impact of the COVID-19 pandemic? | Categorical (nominal), multiple response |
| *Nutrition* | |
| During the COVID-19 pandemic, your household found it easier/no change/more difficult to access healthy and preferred food | Categorical (ordinal), single response |
| How did your household cope with the difficulties in accessing healthy and preferred foods? | Categorical (nominal), multiple response |
| How did your child's consumption of vegetables/staple food/legumes/meat/fish/eggs/dairy change during COVID-19 pandemic? [each food group was asked separately] | Categorical (ordinal), single response |
| *Breastfeeding* | |
| Prior to the COVID-19 pandemic were you breastfeeding? | Yes/no |
| Were you able to continue breastfeeding during the COVID-19 pandemic? | Categorical (ordinal), single response |
| Why were you not able to continue breastfeeding? | Categorical (nominal), multiple response |
| *Growth monitoring* | |
| Before the COVID-19 pandemic, did you monitor your child's growth? | Yes/no |
| During the COVID-19 pandemic, did you monitor your child's growth? | Yes/no |
| Why did you not measure your children during the COVID-19 pandemic? | Categorical (nominal), multiple response |
| *Water and hygiene* | |
| Since COVID-19 pandemic, accessing water has become more difficult/no change/less difficult | Categorical (ordinal), single response |
| Since the COVID-19 pandemic, the amount of water you can access has increased/no change/decreased | Categorical (ordinal), single response |
| If access to water has become more difficult, why? | Categorical (nominal), multiple response |
| Compared to before COVID-19 pandemic, do you wash your hands more often/no change/less often? | Categorical (ordinal), single response |
| Since the COVID-19 pandemic did you experience a change in access to soap? | Categorical (ordinal), single response |

means of a participatory community mapping exercise. Qualitative cut-offs were determined for each of the categories, households classified themselves into one of the socio-economic groups using these cut-offs.

## Data analysis

Data collected from the households through the phone surveys were merged with household demographic information collected in 2017 and 2018 using a unique identifier. Descriptive statistics and standard errors were calculated in Stata version 15. Results are provided for all respondents and stratified by socio-economic group. Correlations with socio-economic status were explored using chi-square tests of which p-values are reported. A correlation is considered statistically significant if the p-value of the chi-square statistics is below 5%.

## Ethical approval

The Bangladesh Medical Research Council provided ethical approval for this study. Informed consent was obtained verbally from all participants.

## Results

### Response rate

The overall response rate was 21%, there is little variation between districts or socio-economic groups, only Barguna (13%) and Jessore (16%) had slightly lower response rates (S1 Table). Main reasons for non-response included disconnected phone numbers and lack of time to participate in the survey.

### Characteristics of study respondents

Most caretakers interviewed for this study were between the ages of 20 and 34 years (77%, n = 314) (Table 2). Eleven percent (n = 45) of caretakers belonged to the poorest socio-economic group, 34% (n = 138) to the poorer socio-economic group, 27% (n = 111) to the middle-income group and 4% (n = 18) to the richest socio-economic group. For 95 respondents (23%) the socio-economic group could not be retrieved because it was not recorded at baseline (n = 2) or due to data-linking issues (n = 93).

The vast majority of caretakers had one child between the ages of 1 and 5 in the household (87%, n = 355). Twelve percent (n = 48) had two children between the ages of 1 and 5 living in the household and only 4 respondents (1%) had three children within that age group living in their household. The youngest child between the ages of 1 and 5 was considered for this study. Sixteen percent (n = 65) of the youngest eligible children were between 12 and 23 months at the time of the survey and 31% of children fell in the oldest age category (48–59 months; n = 128). An approximate equal number of female (50%, n = 202) and male (50%, n = 205) children were included in the study.

### Impact of COVID-19 on livelihood

Table 3 summarizes the results of the survey regarding the impact of COVID-19 on the livelihood of respondents. Eighty-one percent of the respondents perceived it as much more difficult to maintain their livelihoods during the first general lockdown in Bangladesh as compared to before the lockdown. This percentage was higher among the poorest (87%) and poor (88%) respondents as compared to respondents from the middle (75%) and rich (61%) socio-economic groups (p = 0.008). The main reason for this is a reduction in income, which was reported by almost all (95%) respondents irrespective of socio-economic group. Eleven percent of respondents lost their job during the first general lockdown, particularly among those working in the service industry (36%) or day labors (14%). Twenty-three percent of respondents lost business during the first general lockdown (66% of whom reported to work in the business industry). Respondents from the poorest socio-economic group were least likely to report a loss of business (7%), although not statistically significant (p = 0.054) because they were less likely (16%) to work in the business

**Table 2. Characteristics of study respondents.**

|  | N | % |
|---|---|---|
| **Total respondents** | **407** | 100% |
| **Age of respondent** |  |  |
| 18–19 years | 10 | 2% |
| 20–34 years | 314 | 77% |
| 35–49 years | 83 | 20% |
| **Socio-economic group** |  |  |
| Poorest | 45 | 11% |
| Poorer | 138 | 34% |
| Middle | 111 | 27% |
| Rich | 18 | 4% |
| Unknown or unregistered | 95 | 23% |
| **Main source of income**[a] |  |  |
| Agriculture | 105 | 26% |
| Business | 116 | 29% |
| Labour | 153 | 38% |
| Service industry | 59 | 14% |
| Transport | 4 | 1% |
| Outside of Bangladesh | 2 | 0% |
| **Number of children under 5 in household** |  |  |
| One child | 355 | 87% |
| Two children | 48 | 12% |
| Three children | 4 | 1% |
| **Age of youngest eligible child**[b] |  |  |
| 12–23 months | 65 | 16% |
| 24–35 months | 101 | 25% |
| 36–47 months | 113 | 28% |
| 48–59 months | 128 | 31% |
| **Gender of youngest eligible child** |  |  |
| Female | 205 | 50% |
| Male | 202 | 50% |

[a]Respondents could give multiple answers

[b]Caretakers were eligible if they had a child between the ages of 12 and 59 months. The age refers to the youngest child within this age category but caretakers may have children younger than 12 months as well.

industry as compared to poorer (24%), middle-income (37%) and rich (39%) households (p = 0.037). Despite a reduction in income across all socio-economic groups, the poorest were most likely to report not being able to buy food for their households.

In terms of coping mechanisms to deal with the impact of COVID-19 measures on livelihood, most of the respondents had to borrow money (77%) and/or reduce consumption of food and household items (78%). Some of the poorest (14%), poorer (4%) and middle-income (3%) respondents received aid from the government. The complete lockdown (94%) and school closure (39%) were perceived as being most impactful.

## Impact of COVID-19 on breastfeeding practices

Among respondents who were breastfeeding prior to the first general lockdown in Bangladesh 76% continued breastfeeding during the general lockdown, 9% continued breastfeeding to a

**Table 3. Impact of COVID-19 preventive measures on livelihoods, and coping mechanisms.**

| | All (%(SE[a])) | Poorest | Poorer | Middle | Rich | Unknown | χ² p-value |
|---|---|---|---|---|---|---|---|
| **What COVID-19 measures impacted your households the most?** | | | | | | | |
| *Respondents* | *407* | *45* | *138* | *111* | *18* | *95* | |
| Complete lockdown | 94% (1%) | 100% (0%) | 97% (1%) | 88% (3%) | 89% (7%) | 95% (2%) | *P = 0.012* |
| Partial lockdown | 22% (2%) | 11% (5%) | 10% (3%) | 8% (3%) | 6% (6%) | 64% (5%) | *P = 0.000* |
| Canceling social gatherings/events | 0% (0%) | 0% (0%) | 0% (0%) | 1% (1%) | 0% (0%) | 1% (1%) | *P = 0.738* |
| School closure | 39% (2%) | 36% (7%) | 41% (4%) | 42% (5%) | 61% (11%) | 31% (5%) | *P = 0.108* |
| Physical distancing | 13% (2%) | 9% (4%) | 17% (3%) | 18% (4%) | 11% (7%) | 1% (1%) | *P = 0.001* |
| Mandatory mask | 2% (1%) | 2% (2%) | 2% (1%) | 4% (2%) | 6% (6%) | 0% (0%) | *P = 0.396* |
| **Ability to sustain livelihood during COVID-19** | | | | | | | |
| *Respondents* | *407* | *45* | *138* | *111* | *18* | *95* | |
| Much more difficult | 81% (2%) | 87% (5%) | 88% (3%) | 75% (4%) | 61% (11%) | 78% (4%) | *P = 0.008* |
| Somewhat more difficult | 14% (2%) | 9% (4%) | 9% (2%) | 18% (4%) | 28% (11%) | 17% (4%) | *P = 0.083* |
| No change | 5% (1%) | 4% (3%) | 2% (1%) | 7% (2%) | 11% (7%) | 5% (2%) | *P = 0.288* |
| Became easier | 0% (0%) | 0% (0%) | 0% (0%) | 0% (0%) | 0% (0%) | 0% (0%) | - |
| **Impact of COVID-19 measures on livelihood** | | | | | | | |
| *Respondents* | *387* | *43* | *135* | *103* | *16* | *90* | |
| Lost job | 11% (2%) | 12% (5%) | 11% (3%) | 11% (3%) | 13% (8%) | 9% (3%) | *P = 0.980* |
| Lost business | 23% (2%) | 7% (4%) | 21% (4%) | 26% (4%) | 38% (12%) | 26% (5%) | *P = 0.054* |
| Lost key assets | 4% (1%) | 9% (4%) | 6% (2%) | 4% (2%) | 0% (0%) | 1% (1%) | *P = 0.176* |
| Fewer family members able to work | 19% (2%) | 16% (6%) | 26% (4%) | 25% (4%) | 19% (10%) | 1% (1%) | *P = 0.000* |
| Reduced income | 95% (1%) | 100% (0%) | 95% (2%) | 94% (2%) | 94% (6%) | 96% (2%) | *P = 0.626* |
| Not being able to buy/provide food | 15% (2%) | 35% (7%) | 17% (3%) | 10% (3%) | 6% (6%) | 11% (3%) | *P = 0.001* |
| **Coping mechanisms** | | | | | | | |
| Respondents | *387* | *43* | *135* | *103* | *16* | *90* | |
| Sell assets | 25% (2%) | 23% (6%) | 23% (4%) | 23% (4%) | 19% (10%) | 33% (5%) | *P = 0.388* |
| Borrow money | 77% (2%) | 77% (6%) | 81% (3%) | 73% (4%) | 69% (12%) | 76% (5%) | *P = 0.520* |
| Reduce consumption of foods and purchasing of household items | 78% (2%) | 86% (5%) | 76% (4%) | 63% (5%) | 56% (12%) | 97% (2%) | *P = 0.000* |
| Receive aid from government or NGO | 8% (1%) | 14% (5%) | 4% (2%) | 3% (2%) | 0% (0%) | 19% (4%) | *P = 0.000* |
| Other | 3% (1%) | 2% (2%) | 2% (1%) | 5% (2%) | 13% (8%) | 1% (1%) | *P = 0.077* |

[a]SE = Standard Error

lesser extent and 16% discontinued breastfeeding during the lockdown (Fig 2). Fig 2 shows the distribution of (dis)continuation of breastfeeding by age of youngest eligible child on the day of the survey, as well as their estimated age during the first general lockdown. Discontinuation rates are highest among children who were between 38 and 49 months of age during the lockdown (48 to 59 months on the day of the survey), as compared to children who were younger than that during the lockdown (p = 0.000). Respondents of children who were between approximately 2–13 months (12 to 23 months on the day of the survey) and 14–25 (24 to 35 months on the day of the survey) months of age during the first general lockdown were least likely to discontinue breastfeeding, 3% and 9% respectively. The main reason for discontinuation of breastfeeding was that respondents believed it to be not important anymore (82%), potentially because of the child's age, followed by hunger of the mother (17%).

## Access to healthy and preferred foods

Table 4 summarizes the results of the perceived impact of the lockdown on access to food and children's food consumption. Almost all respondents (91%) perceived it to be more difficult to

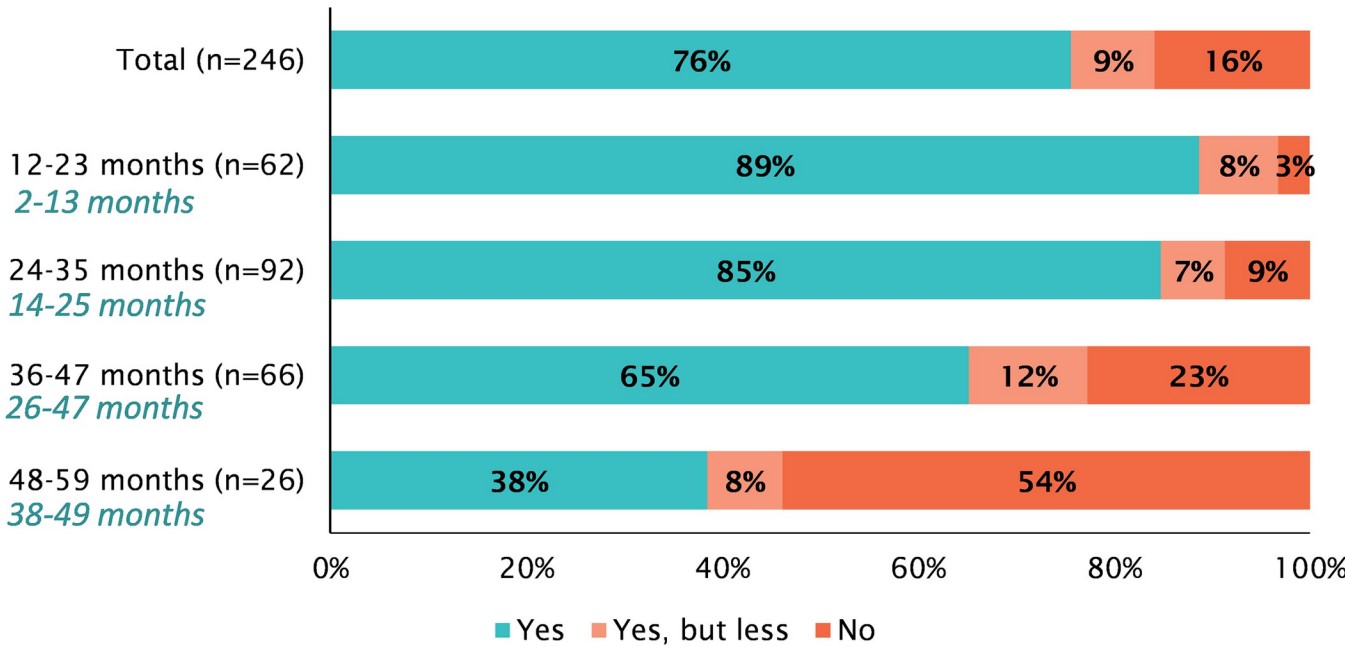

**Fig 2. (Dis)continuation of breastfeeding during COVID-19 by caretakers who were breastfeeding prior to the pandemic by age of child on day of the survey and approximate age at the start of the COVID-19 lockdown.**

access healthy and preferred foods during the general lockdown. To cope with it, 75% of respondents ate less or skipped a meal. There is a clear decreasing trend in the percentage of respondents reporting to skip a meal or eat less from the poorest (79%) to the richest (40%) (p = 0.000). However, the largest share of individuals who reported to eat less or skip a meal were from unknown socio-economic background (96%). Approximately 43% of respondents ate more own-produce food, this was lowest among the poorest respondents (33%) and increased with increasing socio-economic group, though not statistically significant (p = 0.112).

Respondents were asked whether their children reduced consumption of vegetables, staple food (e.g. rice), legumes, meat, fish, eggs or dairy during the lockdown. Reductions were reported across food groups and socio-economic groups. However, reduction of consumption of animal protein sources such as meat (88%), fish (83%), eggs (76%) and dairy (85%) were most frequently reported. According to 77% of respondents, children reduced consumption of at least 4 food groups. This was higher among the poorest (84%) and poorer respondents (81%), than among middle-income (65%) and rich respondents (67%) (p = 0.003). Only 31% of respondents reported an increase in at least one food group, primarily of legumes (23%) and or vegetables (14%).

### Growth monitoring

Prior to the pandemic 86% of respondents reportedly monitored the growth of their children (Table 5). Among these respondents, 44% continued monitoring their child's growth during the pandemic, ranging from 66% among the poorest respondents to 25% among the richest respondents (p = 0.004). Main reasons for discontinuation of growth monitoring were that the service was not available (73%) and fear of infection (41%). Discontinuation of growth monitoring out of fear of infection was more often reported by respondents who received these services from a formal health provider (60%; e.g. health complex, community clinic or private

**Table 4. Impact of COVID-19 on access to healthy and preferred foods, and consumption of different foods by children.**

| | All | Poorest | Poorer | Middle | Rich | Unknown | χ² p-value |
|---|---|---|---|---|---|---|---|
| **Access to healthy and preferred food** | | | | | | | |
| *Respondents* | *407* | *45* | *138* | *111* | *18* | *62* | |
| Easier | 1% (0%) | 0% (0%) | 1% (1%) | 2% (1%) | 6% (6%) | 0% (0%) | P = 0.335 |
| No change | 8% (1%) | 7% (4%) | 7% (2%) | 11% (3%) | 11% (7%) | 6% (3%) | P = 0.707 |
| More difficult | 91% (1%) | 93% (4%) | 91% (2%) | 87% (3%) | 83% (9%) | 94% (3%) | P = 0.386 |
| **Coping strategies for reduced access to healthy and preferred foods** | | | | | | | |
| *Respondents* | *370* | *42* | *126* | *91* | *15* | *90* | |
| Reduce household expenses | 25% (2%) | 19% (6%) | 13% (3%) | 11% (3%) | 7% (7%) | 63% (5%) | P = 0.000 |
| Eat more own-produced food | 43% (3%) | 33% (7%) | 38% (4%) | 52% (5%) | 60% (13%) | 43% (5%) | P = 0.112 |
| Eat less or skip meal | 75% (2%) | 79% (6%) | 71% (4%) | 65% (5%) | 40% (13%) | 96% (2%) | P = 0.000 |
| Buy cheaper food | 38% (3%) | 31% (7%) | 35% (4%) | 28% (5%) | 20% (10%) | 61% (5%) | P = 0.000 |
| Borrow food from relatives | 12% (2%) | 12% (5%) | 13% (3%) | 8% (3%) | 7% (7%) | 14% (4%) | P = 0.685 |
| Borrow money | 73% (2%) | 76% (7%) | 76% (4%) | 66% (5%) | 67% (12%) | 74% (5%) | P = 0.455 |
| Received food aid | 4% (1%) | 7% (4%) | 3% (2%) | 0% (0%) | 0% (0%) | 10% (3%) | P = 0.010 |
| Other | 2% (1%) | 2% (2%) | 1% (1%) | 3% (2%) | 7% (7%) | 2% (1%) | P = 0.564 |
| **Decreased consumption of food groups by young children** | | | | | | | |
| *Respondents* | *407* | *45* | *138* | *111* | *18* | *95* | |
| Vegetables | 32% (2%) | 31% (7%) | 25% (4%) | 20% (4%) | 28% (11%) | 56% (5%) | P = 0.000 |
| Staple food | 34% (2%) | 36% (7%) | 26% (4%) | 22% (4%) | 17% (9%) | 65% (5%) | P = 0.000 |
| Legumes | 25% (2%) | 27% (7%) | 25% (4%) | 23% (4%) | 22% (10%) | 27% (5%) | P = 0.973 |
| Meat | 88% (2%) | 89% (5%) | 91% (2%) | 81% (4%) | 78% (10%) | 93% (3%) | P = 0.025 |
| Fish | 83% (2%) | 87% (5%) | 88% (3%) | 75% (4%) | 78% (10%) | 83% (4%) | P = 0.046 |
| Eggs | 76% (2%) | 82% (6%) | 82% (3%) | 67% (4%) | 61% (11%) | 80% (4%) | P = 0.023 |
| Dairy | 85% (2%) | 91% (4%) | 91% (2%) | 78% (4%) | 67% (11%) | 85% (4%) | P = 0.006 |
| At least 4 food groups | 77% (2%) | 84% (5%) | 81% (3%) | 65% (5%) | 67% (11%) | 84% (4%) | P = 0.003 |

**Table 5. Discontinuation of growth monitoring services and reasons.**

| | All | Poorest | Poorer | Middle | Rich | Unknown | χ² p-value |
|---|---|---|---|---|---|---|---|
| **Growth monitoring prior to pandemic** | | | | | | | |
| *Respondents* | *407* | *45* | *138* | *111* | *18* | *95* | |
| Yes | 86% (2%) | 91% (4%) | 81% (3%) | 83% (4%) | 89% (7%) | 94% (2%) | P = 0.053 |
| No | 14% (2%) | 9% (4%) | 19% (3%) | 17% (4%) | 11% (7%) | 6% (2%) | |
| **Continuation of growth monitoring during pandemic** | | | | | | | |
| *Respondents* | *350* | *41* | *112* | *92* | *16* | *89* | |
| Yes | 44% (3%) | 66% (7%) | 44% (5%) | 33% (5%) | 25% (11%) | 48% (5%) | P = 0.004 |
| No | 56% (3%) | 34% (7%) | 56% (5%) | 67% (5%) | 75% (11%) | 52% (5%) | |
| **Reasons for discontinuation of growth monitoring** | | | | | | | |
| *Respondents* | *196* | *14* | *63* | *61* | *12* | *46* | |
| Services were not available | 73% (3%) | 79% (11%) | 81% (5%) | 66% (6%) | 67% (14%) | 72% (7%) | P = 0.376 |
| Fear of infection | 41% (4%) | 29% (12%) | 44% (6%) | 56% (6%) | 25% (13%) | 26% (6%) | P = 0.016 |
| COVID-19 restrictions | 2% (1%) | 7% (7%) | 0% (0%) | 2% (2%) | 8% (8%) | 0% (0%) | P = 0.081 |
| No transportation available | 1% (1%) | 0% (0%) | 0% (0%) | 2% (2%) | 0% (0%) | 0% (0%) | P = 0.695 |
| General costs | 1% (1%) | 0% (0%) | 0% (0%) | 2% (2%) | 0% (0%) | 2% (2%) | P = 0.695 |
| Could not afford transport | 1% (1%) | 0% (0%) | 0% (0%) | 0% (0%) | 0% (0%) | 0% (0%) | P = 0.512 |

clinic) as compared to respondents who received these services through NGO or NGO affiliated providers (37%) (p = 0.010). The latter were more likely to discontinue growth services due to lack of availability of services during the pandemic (81%) as compared to those receiving these services from a formal provider (33%) (p = 0.000) (Table 5).

### Access to water and soap

The majority of respondents experienced no change in the ease of obtaining water (67%) or its quantity (51%) during the general lockdown. In fact, many respondents reported an increase in the quantity of water they could access (41%). Among those who experienced more difficulties in accessing water, main reasons included increased distance to water source (46%), longer waiting times (61%) and movement restrictions (36%). Almost all respondents reported that their frequency of handwashing had increased (99%), and that soap was more easily accessible during the lockdown (99%). Finally, all caretakers with a child between the ages of 1 and 5 years, who experienced diarrhea during the pandemic (12%) accessed health services for their child (Table 6).

## Discussion
### Impact of COVID-19 on livelihoods, nutrition, water and hygiene

This study assessed the impact of the first general lockdown in Bangladesh, on the livelihood, access to water, soap and health services and nutrition of young children in rural Bangladesh.

**Table 6. Perceived changes in access to water, soap, hand washing frequency and reasons for difficulties in accessing water.**

| | All | Poorest | Poorer | Middle | Rich | Unknown | $\chi^2$ p-value |
|---|---|---|---|---|---|---|---|
| **Changes in access to water** | | | | | | | |
| *Respondents* | *407* | *45* | *138* | *111* | *18* | *95* | |
| More difficult | 14% (2%) | 13% (5%) | 17% (3%) | 18% (4%) | 22% (10%) | 6% (2%) | *P = 0.104* |
| No change | 67% (2%) | 71% (7%) | 64% (4%) | 52% (5%) | 56% (12%) | 89% (3%) | *P = 0.000* |
| Less difficult | 18% (2%) | 16% (5%) | 20% (3%) | 30% (4%) | 22% (10%) | 4% (2%) | *P = 0.000* |
| **Changes in quantity of water** | | | | | | | |
| *Respondents* | *407* | *45* | *138* | *111* | *18* | *95* | |
| Increased | 41% (2%) | 16% (5%) | 30% (4%) | 30% (4%) | 17% (9%) | 88% (3%) | *P = 0.000* |
| No change | 51% (2%) | 76% (6%) | 65% (4%) | 59% (5%) | 61% (11%) | 8% (3%) | *P = 0.000* |
| Decreased | 8% (1%) | 9% (4%) | 5% (2%) | 12% (3%) | 22% (10%) | 3% (2%) | *P = 0.016* |
| **Reasons for difficulties in accessing water** | | | | | | | |
| *Respondents* | *59* | *6* | *23* | *20* | *4* | *6* | |
| Increased distance | 46% (6%) | 50% (20%) | 39% (10%) | 50% (11%) | 75% (22%) | 33% (19%) | *P = 0.672* |
| Longer waiting times | 61% (6%) | 67% (19%) | 57% (10%) | 70% (10%) | 75% (22%) | 33% (19%) | *P = 0.522* |
| Reduced availability | 2% (2%) | 0% (0%) | 0% (0%) | 0% (0%) | 0% (0%) | 17% (15%) | *P = 0.061* |
| Technical problems with source | 3% (2%) | 0% (0%) | 0% (0%) | 5% (5%) | 0% (0%) | 17% (15%) | *P = 0.337* |
| Movement restrictions | 36% (6%) | 50% (20%) | 43% (10%) | 25% (10%) | 50% (25%) | 17% (15%) | *P = 0.486* |
| **Changes in hand washing frequency** | | | | | | | |
| *Respondents* | *407* | *45* | *138* | *111* | *18* | *95* | |
| More often | 99% (0%) | 98% (2%) | 99% (1%) | 99% (1%) | 100% (0%) | 100% (0%) | *P = 0.709* |
| No change | 1% (0%) | 2% (2%) | 1% (1%) | 1% (1%) | 0% (0%) | 0% (0%) | *P = 0.709* |
| Less often | 0% (0%) | 0% (0%) | 0% (0%) | 0% (0%) | 0% (0%) | 0% (0%) | - |
| **Changes in availability of soap** | | | | | | | |
| *Respondents* | *407* | *45* | *138* | *111* | *18* | *95* | |
| More access | 99% (0%) | 98% (2%) | 99% (1%) | 99% (1%) | 100% (0%) | 100% (0%) | *P = 0.692* |
| No change | 1% (0%) | 2% (2%) | 1% (1%) | 1% (1%) | 0% (0%) | 0% (0%) | *P = 0.692* |
| Less access | 0% (0%) | 0% (0%) | 0% (0%) | 0% (0%) | 0% (0%) | 0% (0%) | - |

The lockdown came into effect on March 26[th] and lasted for 65 days. During this period all non-essential businesses were closed.

In line with other studies [19], almost all respondents (95%) across all socio-economic groups experienced a reduction in income during the COVID-19 pandemic, but poor and poorer households were disproportionally affected by the consequences of reduced income. To cope with this, many households resorted to borrowing money from friends or family (77%). However, it is the poorest households which were more likely to report not being able to provide ample food and having to reduce consumption of food and household items (poorest: 86%, poorer:76%, middle-income: 63%, rich: 56%).

The results of this study suggest that diet diversity and food intake of young children were severely impacted by the pandemic, increasing the risk of malnutrition. A Bangladesh National Nutrition Council (BNNC) report outlining the potential impact of COVID-19 pandemic warned of increased prices for staple foods such as rice and pulses, as well as almost all types of meat (7). In combination with the reduced purchasing power of families, it is not surprising that consumption of many food groups decreased. Three out of four caretakers reported a decrease in consumption of at least 4 different food groups for young children. Particularly animal protein sources such as meat, fish, dairy and eggs were reduced in the diets of children. There was little compensation from other food groups as only one third of caretakers reported an increase in at least one food group. Reductions in consumption were reported by all socio-economic groups, but were highest among the poorest and poorer caretakers for all food groups. This is particularly concerning as the poorest are disproportionally affected by malnutrition [20, 21]. As such, there is a valid concern that existing inequalities in malnutrition and related health outcomes between socio-economic groups are exacerbated.

Fortunately, breastfeeding practices appeared not to be impacted by COVID-19 as discontinuation was low among 2–25 months old children. This finding suggests that the dietary impact for children who were exclusively breastfed during the first lockdown is not as severe as for children receiving complementary feeding. Initiation and duration of breastfeeding could have been negatively impacted due to reduced access to counseling and education from health care professionals (i.e. during ante- and postnatal care), fear or confusion around safety of breastfeeding while sick or inability to breastfeed due to sickness [22–24].

Recurrent diarrhea in young children is a known risk factor for stunting in children under 2 years of age in Bangladesh and should be appropriately managed [25]. Factors increasing the risk of diarrhea in Bangladesh are related to improper hygiene and lack of access to safe water and improved sanitation [26, 27]. Twelve percent of caretakers reported that their child had experienced diarrhea since the start of the COVID-19 pandemic. The incidence of diarrhea observed for the time period is not concerning, considering the 2-week incidence of 5% reported by the DHS in 2018. If diarrhea is a proxy for hygiene and sanitation, the reported prevalence does not cause any reason for concern. In addition, the impact of COVID-19 on access to water seems to be limited, with only very few respondents experiencing reduced access to water due to movement restrictions and longer waiting times. In addition, in line with other studies self-reported handwashing and access to soap have increased during the COVID-19 pandemic [28–30]. Although these results are prone to social desirability bias, it is likely that awareness of the importance of hand washing has increased due to awareness raising campaigns of the government and NGOs [30].

National statistics on health care utilization show a downward trend in attendance of community clinics by children and mothers. In addition, routine immunization activities were limited, and sporadic outbreaks of vaccine preventable diseases were reported in parts of Bangladesh [31–33]. Disruptions could be related to restricted mobility, fear of infection, financial barriers and limited capacity of the health care system [34–36]. Utilization of health

services was measured in two ways by this study: (1) health seeking for children with diarrhea, and (2) continuation of growth monitoring services for children. Fortunately, results from our study show that all caretakers of children who had diarrhea at any point during the pandemic sought treatment for their child. However, utilization of growth monitoring services was discontinued by 56% of respondents who used them prior to the pandemic. Respondents who accessed growth monitoring services from formal health care providers prior to the start of the pandemic discontinued the monitoring out of fear of infection, while respondents using the services from NGOs or other non-formal care providers could not continue monitoring their child's growth as services were not available. The impact of COVID-19 on health seeking behavior might be different for curative services as compared to preventive services, and disruption of NGO activities contributed to limited access to health services. A new Demographic and Health Survey is planned for 2022, future research should utilize these data to quantify the impact of COVID-19 on adverse health outcomes, malnutrition, and health seeking behavior for different subpopulations in more detail.

## Mitigation strategies to limit impact of COVID-19, and the role of NGOs

Only eight percent of households mentioned that they received aid from the government during the first general lockdown, this was slightly higher among the poorest households (14%) but still very low. During the first wave of COVID-19, the government of Bangladesh implemented a number of social safety net programs including food and cash aid [17]. However, as a result of financial constraints the coverage of these SSNPs appeared insufficient and the duration too short. One issue identified in the implementation of the SSNPs was the lack of information systems. As such government officials spend precious time identifying, registering and verifying potential beneficiaries [17]. NGOs often collect data to monitor their progress against targets and for accountability toward donors. As such, they have a wealth of information on the communities they work with, which could be leveraged by governments to speed up the identification of potential beneficiaries. In addition, these data can be used by governments, NGOs and international actors to design more effective programs to support the communities.

The results suggest that respondents were aware of the importance of handwashing, likely due to awareness raising activities by government and NGOs. The government of Bangladesh rolled out an extensive awareness campaign via social media, radio and other media. A study on how risk communication was received revealed that although the campaign appears to have reached most of the population, the information was not always perceived to be adequate [37]. Understanding of the risk communication on COVID-19 treatment, vaccination and diagnostic tests was inadequate and low educated respondents in particular lacked understanding of the COVID-19 information [37]. NGOs have long-term, well-established and strong relationships with communities, and governments should be encouraged to leverage these trust relationships and encourage them to use their network to reach communities with vital information and services more directly and in clear language. Governments can also play a coordinating role, by dividing tasks and regions that use the strength of different NGOs to reach target populations in the quickest and most efficient manner possible. Especially where NGOs have contact details of community leaders, these should be shared with governments, when consent is given, to enhance community engagement and messaging.

Long term investments are needed to enhance the resilience of the communities and efficiency of the government. Stronger SSNPs including protection of informal workers, and enhanced data infrastructure are needed to respond to shocks–including but not limited to the COVID-19 pandemic–in an appropriate and time-efficient manner. In the meantime, to

minimize the need for lockdowns as a measure to stop community spread of COVID-19, the international community has to ensure that sufficient amounts of vaccines are made available to all low- and middle-income countries.

## Strengths and limitations

Due to a data management issue, household unique identifiers of 95 households could not be linked to an existing baseline dataset which has information on household demographics. For these households, information on socio-economic group could not be retrieved. All of these households fell under one particular partner NGO.

The sampling frame used for to draw the sample of households was mostly based on data collected in 2017 and 2018 updated with information collected in 2019, it is therefore unclear to what extent this sampling frame covers all households with children under the age of 5 and to what extent phone numbers were still active. Furthermore, even though data on socio-economic group was collected 3 years before this study, they provide a good overview of the long-term socio-economic group of a household and is not affected by shocks that may have occurred in late 2019 or early 2020 right before the COVID-19 lockdown.

The response rate was low (21%) due to a number of issues including disconnected phone numbers and lack of time to participate, this may lead to bias in the results if the response rate is different across different communities. Fortunately, there was little variation in response rates between PNGOs and socio-economic groups.

Despite the low response rate, the database provided a unique opportunity to perform simple random sampling which is often not possible in population studies due to lack of a sampling frame. Usually, a clustered sampling approach is needed to circumvent the lack of a sampling frame, which results in a higher sample size to meet the same effective sample size under simple random sampling [38].

Finally, the results are based on perceived and self-reported changes and not observed differences, making them prone to social desirability bias. Nonetheless, they provide relevant insights into the lived experiences of these communities and the challenges they faced during the pandemic. A demographic and health survey is planned for 2022 in Bangladesh, this will provide further insights into the impact of the pandemic on health outcomes and sociodemographic indicators.

## Conclusions

Bangladesh has been hit hard by the pandemic, the government's response to contain COVID-19 had strong negative impact on rural households with young children. Due to loss of income, maintaining livelihood became more difficult for many. This also impacted the quality and quantity of the diets of young children, in particular the consumption of animal source foods were reduced with little to no compensation of other food groups. The short term and long-term consequences of this on the nutritional status of young children can be disastrous and could set back years of progress in the fight against malnutrition. While economic hardship was experienced by households from all socio-economic groups, the poorest and poor were more likely to report a decrease in consumption with the risk of exacerbating existing health inequities.

Restrictive measures like lockdowns to limit the spread of COVID-19 may still be necessary in the future. Without strong social safety net programs vulnerable populations will continue to be disproportionally affected by the economic hardship imposed on them. The implementation of social safety net programs by the government can be sped up by active engagement of NGOs, building on their data and information systems to strengthen their own. NGOs' well-

established relationships with communities can be leveraged to more effectively deliver critical messages. Long-term investments in social safety net programs, nutrition and WASH programs are required to build more resilient communities and undo damage done by the pandemic.

## Supporting information

**S1 Table. Response rate by district and socio-economic group.**
(DOCX)

**S1 File. Questionnaire on inclusivity in global research.**
(DOCX)

## Acknowledgments

We would like to acknowledge the work of Masudul Alam and Hafizur Rahman in providing essential input in the design, coordination and supervision of the data collection, as well as the data collectors for conducting the interviews in a diligent manner. Their role was essential to complete this study. We also thank all participating respondents for taking the time to share their experiences.

## Author Contributions

**Conceptualization:** Margo van Gurp, Imam M. Riad, Kazal A. Islam, Remco M. Geervliet, Mirjam I. Bakker.

**Data curation:** Imam M. Riad, Kazal A. Islam.

**Formal analysis:** Margo van Gurp.

**Methodology:** Margo van Gurp, Md Shariful Islam, Mirjam I. Bakker.

**Supervision:** Mirjam I. Bakker.

**Validation:** Md Shariful Islam.

**Writing – original draft:** Margo van Gurp.

**Writing – review & editing:** Imam M. Riad, Kazal A. Islam, Md Shariful Islam, Remco M. Geervliet, Mirjam I. Bakker.

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
