## [Decision Letter · Decision Letter 0]

10 Aug 2022

PONE-D-22-19605WASH, nutrition and health-seeking behavior during COVID-19 lockdowns: evidence from rural BangladeshPLOS ONE

Dear Dr. van Gurp,

Thank you for submitting your manuscript to PLOS ONE. After careful consideration, we feel that it has merit but does not fully meet PLOS ONE’s publication criteria as it currently stands. Therefore, we invite you to submit a revised version of the manuscript that addresses the points raised during the review process.

We look forward to receiving your revised manuscript.

Kind regards,

Larissa Loures Mendes, Ph.D.

Academic Editor

PLOS ONE

Journal Requirements:

Additional Editor Comments:

This study aimed understand how livelihood, access to water, food consumption of young children and health seeking behavior in communities in rural Bangladesh have been affected by the first general lockdown imposed between March 26 and May 30th 2020. The manuscript is interesting, but it needs minor revision. The Reviewers have provided feedback for the authors to improve this work. Below are specific comments.

Reviewer 1:

Comments to the Author

Thanks to authors for presenting a study on very relevant subject: assessing impact of COVID-19 on socioeconomic activities, health, nutrition and WASH. However, in the result section (line 201-216) it is difficult to fathom the impact of COVID-19 on breastfeeding practices. It would be good to elaborate in discussion section how the authors believe COVID-19 had any influenced breastfeeding practices. Further, in the same result section how did the author capture information from parents with children of age group 2-13 months, when it was not in sampling frame as per methodology. Kindly address or explain. Many thanks again

Reviewer 2:

Comments to the Author

The article presented is interesting, however, here are some suggestions:

1. The study hypothesis is not clear in the introduction.

2. It would be interesting to better describe the data collection instrument. Information is lacking to understand the variables.

3. In the data analyses, describe which was the dependent variable and which were the independent ones.

4. Perform a multiple analysis to adjust for possible confounding variables.

5. Why were the associations made only with the variable "Socio-economic group"?

Reviewers' comments:

Reviewer's Responses to Questions

**Comments to the Author**

1. Is the manuscript technically sound, and do the data support the conclusions?

Reviewer #1: Yes

Reviewer #2: No

2. Has the statistical analysis been performed appropriately and rigorously? 

Reviewer #1: Yes

Reviewer #2: No

3. Have the authors made all data underlying the findings in their manuscript fully available?

Reviewer #1: Yes

Reviewer #2: Yes

4. Is the manuscript presented in an intelligible fashion and written in standard English?

Reviewer #1: Yes

Reviewer #2: Yes

5. Review Comments to the Author

Reviewer #1: Thanks to authors for presenting a study on very relevant subject: assessing impact of COVID-19 on socioeconomic activities, health, nutrition and WASH. However, in the result section (line 201-216) it is difficult to fathom the impact of COVID-19 on breastfeeding practices. It would be good to elaborate in discussion section how the authors believe COVID-19 had any influenced breastfeeding practices. Further, in the same result section how did the author capture information from parents with children of age group 2-13 months, when it was not in sampling frame as per methodology. Kindly address or explain.

Many thanks again

Reviewer #2: The article presented is interesting, however, here are some suggestions:

1. The study hypothesis is not clear in the introduction.

2. It would be interesting to better describe the data collection instrument. Information is lacking to understand the variables.

3. In the data analyses, describe which was the dependent variable and which were the independent ones.

4. Perform a multiple analysis to adjust for possible confounding variables.

5. Why were the associations made only with the variable "Socio-economic group"?

6. PLOS authors have the option to publish the peer review history of their article (what does this mean?). If published, this will include your full peer review and any attached files.

Reviewer #1: **Yes: **Shahwar Kazmi

Reviewer #2: No

---

## [Author Response · Author response to Decision Letter 0]

7 Nov 2022

Dear Reviewers,

Thank you very much for your carefully considered review of our article titled ‘WASH, nutrition and health-seeking behavior during COVID-19 lockdowns: evidence from rural Bangladesh’. We have responded to each of your comments in the table below, and made a majority of the suggested revisions. All changes can be found in the version with the ‘_Track changes’ suffix. 

We hope that the updated version satisfies the reviewers’ concerns, and look forward to your feedback if additional changes are requested. 

With kind regards,

Margo van Gurp

KIT Royal Tropical Institute

Response to reviewer 1

Comment 1: Thanks to authors for presenting a study on very relevant subject: assessing impact of COVID-19 on socioeconomic activities, health, nutrition and WASH

Response 1: Thank you for your kind feedback and for your suggestions to improve this paper.

Comment 2: However, in the result section (line 201-216) it is difficult to fathom the impact of COVID-19 on breastfeeding practices. It would be good to elaborate in discussion section how the authors believe COVID-19 had any influenced breastfeeding practices

Response 2: Thank you for your comment, indeed the relation between breastfeeding practices and covid-19 may not be evident. We have elaborated on how breastfeeding practices could have been affected by the pandemic in lines 306 - 311 in the discussion.

Comment 3: Further, in the same result section how did the author capture information from parents with children of age group 2-13 months, when it was not in sampling frame as per methodology.

Response 3: The study was carried out approximately 10 months after the first general lockdown. The age group of 2-13 months you are referring to is the 'estimated age of the child during the lockdown' which is different from the age at time of the survey. While this is mentioned in the text, we agree with the reviewer that it could be beneficial to make this more evident. As such we indicated the age of the child at the time of the survey in text too (lines 218-221), and in the methods section (lines 147 - 148).

Response to reviewer 2

Comment 1: The study hypothesis is not clear in the introduction.

Response 1: Thank you for your review and valuable suggestion for improvement. We elaborated on our hypothesis in the introduction (lines 90 - 96)

Comment 2: It would be interesting to better describe the data collection instrument. Information is lacking to understand the variables.

Response 2: We have included a table highlighting the questions from the survey that were analyzed as part of this study, we hope that this provides the relevant details (Table 1).

Comment 3: In the data analyses, describe which was the dependent variable and which were the independent ones.

Response 3: The terminology 'dependent' versus 'independent' variable is specific to regression analysis, where typically one or more variables of interest (independent variables) are explored for their association with one outcome (dependent variable). However, this analysis does not describe a regression analysis but a simple test of 'independence' between two variables by means of a Chi-square test. A chi square does not distinguish between a 'dependent' and 'independent' variable. We have added an overview of key outcome variables described in this study in Table 1 of the methods section, we hope this makes it more clear.

Comment 4: Perform a multiple analysis to adjust for possible confounding variables.

Response 4: Thank you for your comment and we appreciate the effort to bring the analysis to a higher level. However, we do not believe that a multivariable analysis is in line with the scope of this study. The aim of the analysis is mainly descriptive in nature and describes many potential 'outcome variables'. We believe that a regression model (or other statistical models that allow for the adjustment of confounding and effect modification) is more appropriate when there is one or a few clearly defined outcome measures, and researchers are looking for determinants that are associated with that (set of) outcome(s). Datasets such as Demographic and Health Surveys are more suitable for this type of research.

Comment 5: Why were the associations made only with the variable "Socio-economic group"?

Response 5: We have stated the objectives of this study more clearly in the introduction (87 - 90), we hope that this addresses your question.

Journal requirements

Response: we have carefully reviewed the documents and have formatted accordingly.

Response: We have included said questionnaire as supporting information 2

3. PLOS requires an ORCID iD for the corresponding author in Editorial Manager on papers submitted after December 6th, 2016. Please ensure that you have an ORCID iD and that it is validated in Editorial Manager. To do this, go to ‘Update my Information’ (in the upper left-hand corner of the main menu), and click on the Fetch/Validate link next to the ORCID field. This will take you to the ORCID site and allow you to create a new iD or authenticate a pre-existing iD in Editorial Manager. Please see the following video for instructions on linking an ORCID iD to your Editorial Manager account:

Response: ORCID account is provided.

4. We note that Figure 1 in your submission contain [map/satellite] images which may be copyrighted. All PLOS content is published under the Creative Commons Attribution License (CC BY 4.0), which means that the manuscript, images, and Supporting Information files will be freely available online, and any third party is permitted to access, download, copy, distribute, and use these materials in any way, even commercially, with proper attribution. For these reasons, we cannot publish previously copyrighted maps or satellite images created using proprietary data, such as Google software (Google Maps, Street View, and Earth). For more information, see our copyright guidelines: http://journals.plos.org/plosone/s/licenses-and-copyright

Response: the figure is not copyrighted but created by the lead author.

Response: the reference list has been reviewed.

---

## [Editor Report · Decision Letter 1]

18 Nov 2022

WASH, nutrition and health-seeking behavior during COVID-19 lockdowns: evidence from rural Bangladesh

PONE-D-22-19605R1

Dear Dr. van Gurp,

I appreciate the careful review and clarification of the issues addressed by the reviewers, after adjustments the manuscript became even more interesting for publication.

We’re pleased to inform you that your manuscript has been judged scientifically suitable for publication and will be formally accepted for publication once it meets all outstanding technical requirements.

Kind regards,

Larissa Loures Mendes, Ph.D.

Academic Editor

PLOS ONE

---

## [Editor Report · Acceptance letter]

29 Nov 2022

PONE-D-22-19605R1 

WASH, nutrition and health-seeking behavior during COVID-19 lockdowns: evidence from rural Bangladesh 

Dear Dr. van Gurp:

I'm pleased to inform you that your manuscript has been deemed suitable for publication in PLOS ONE. Congratulations! Your manuscript is now with our production department. 

Kind regards, 

on behalf of

Dr. Larissa Loures Mendes 

Academic Editor

PLOS ONE